# Investigating the Link between Early Life and Breast Anomalies

**DOI:** 10.3390/children10030601

**Published:** 2023-03-21

**Authors:** Panagiotis Christopoulos, Alkis Matsas, Makarios Eleftheriades, Georgia Kotsira, Anna Eleftheriades, Nikolaos F. Vlahos

**Affiliations:** Second Department of Obstetrics and Gynecology, “Aretaieion” Hospital, Faculty of Medicine, National and Kapodistrian University of Athens, 11528 Athens, Greece

**Keywords:** breast, risk factors, anomalies, pathology, childhood, adolescence

## Abstract

Several factors during childhood and adolescence are thought to be associated with the development of proliferative benign breast diseases and breast cancer in adulthood. In order to identify them, the authors conducted an extensive review of the literature up to October 2022, searching for clinical studies, reports, and guidelines in English. A thorough Medline/Pubmed and Google scholar database research was performed, investigating the link between diet, exercise, age of menarche, body mass index, ionizing radiation exposure during childhood and adolescence, and proliferative breast diseases and breast cancer in adulthood. A list of keywords, including breast disorders, adolescence, childhood, and breast cancer was included in our search algorithm. Numerous studies concede that the development of breast disease in adulthood is influenced by various risk factors, whose influence begins during early childhood and adolescence.

## 1. Introduction

Numerous studies have tried to explore the relationship between early life exposures and the risk for benign breast disease (BBD). Relevant literature has shown that proliferative BBD may also increase the subsequent risk for breast cancer [1]. Understanding the influence of these factors and life events during childhood and adolescence on the development of BBD can provide information regarding the pathophysiology of breast cancer and help develop prevention strategies [2].

Some of the earliest epidemiological studies highlighted the role of factors such as age at menarche onset, age at first birth, total body fat, and adolescent body mass index (BMI) in determining subsequent risk of breast cancer [3]. These observations suggested that breast tissue may be vulnerable during the time between the onset of menarche, when the breast cells begin to proliferate, and the completion of the first pregnancy, when breast tissue undergoes terminal differentiation into milk-producing cells [1].

## 2. Materials and Methods

In order to identify factors acting during childhood and adolescence and are potentially related with the development of proliferative benign breast diseases and breast cancer in adulthood, the authors conducted an extensive Medline/Pubmed and Google scholar database search of the literaturein English up until October 2022. A list of keywords including breast disorders, adolescence, childhood, and breast cancer was included in our search algorithm.

## 3. Results

### 3.1. Dietary Factors

In 1988 deWaad and Trichopoulos proposed that an energy-rich diet during puberty and adolescence stimulates the growth of mammary glands and leads to an increased occurrence of precancerous breast lesions [4]. Furthermore, diet alters the hormonal environment of the breast [5].

In a cohort study during adolescence, Baer et al. examined the relation between type of fat and BBD. Those who were in the top quintile of animal fat consumption had a 33% increased risk of proliferative breast diseases (P-BBD), whereas women who were in the highest quintile of vegetable fat consumption had a 27% reduced risk of P-BBD. The highest quintile of monounsaturated fat consumption was associated with a relative risk of 1.52. No association with total fat consumption was found [6].

As far as meat consumption is concerned, several hypotheses explain how intake of red meat could induce carcinogenesis: its highly bioavailable iron content, growth- promoting hormones, carcinogenic heterocyclic amines formed in cooking, and fatty acids contained in the meat [7].

In a prospective cohort study, Linos et al. found that women who consumed the highest amounts of red meat presented an elevated risk of breast cancer (RR, 1.34; 95% CL, 0.94–1.89; *p* Value = 0.05) compared with those at the lowest quintile. In fact, the association was stronger for negative estrogen receptor (ER−) and progesterone (PR+) positive tumors [7].

Another study by Farvid et al. reported that total fruit consumption during adolescence was associated with a lower risk of breast cancer. In fact, participants with a median intake of 2.9 servings per day seemed to have a significantly lower risk than those whose median intake was 0.5 servings, with a hazard ratio (HR) of 0.75. The association for fruit intake during adolescence was independent of adult fruit intake [8]. Women in the highest quintile of adolescent fiber intake had a 25% lower risk of P-BBD compared to women in the lowest quintile [9]. A study by Liu et al. showed an inverse relation between invasive breast cancer and the consumption of dietary fiber and vegetable protein in adolescence [10].

Several biological mechanisms could explain the protective effect of fiber consumption from breast tumors. Dietary fiber may increase excretion of estrogen by inhibiting the reabsorption of estrogen from the gastrointestinal tract. Additionally, the protective effect may also be partly due to the anti-estrogenic effects of lignans, which have inhibitory effect on cell proliferation in breast tumors [9].

Soy intake during childhood and adolescents as well as its properties have been thoroughly examined. A retrospective cohort of Asian migrants to the USA demonstrated the protective effect of higher soy intake during childhood (OR 0.40). This protective effect was found to be weaker during the adolescent (0.80) and adult years (OR 0.75) [11].

A meta-analysis including seven studies concluded that soy intake and breast cancer risk are variables that seem to be inversely related. More specifically, the meta-analysis reported that among Asian females, the ones who received a daily 20 mg dose of isoflavone had a decreased likelihood of developing breast cancer, when compared to the females that received a 5 mg daily dose [12]. This protective association was further studied and confirmed by a subsequent prospective study in Shanghai [13].

In a prospective cohort study of 9031 females by Berkey et al., it was found that biopsy-confirmed BBD can be associated with certain adolescent dietary factors. In particular, the authors suggested that the consumption of animal (nondairy) fat at 10 years of age was associated with a higher risk for BBD, while the consumption of nuts and peanut butter at 14 years of age was associated with a lower risk for the development of the disease. Both are individually important milestones as they represent dietary exposures before and after adolescent height growth, which is typically completed by 14 years of age [14].

### 3.2. Lifestyle Factors

#### 3.2.1. Alcohol

According to the International Agency for Research on Cancer, alcohol is closely linked with invasive breast carcinoma [15]. Relatively few studies have examined the effect of alcohol consumption among young people and adolescents regarding the risk of breast cancer [16,17].

A meta-analysis thatwas published in 2017 and based on 20 prospective studies found an association between heavy alcohol intake and ER+ breast cancer (≥30 g per day of alcohol consumption versus nondrinkers, RR = 1.35 (1.23–1.48)). However, the association between alcohol and ER–breast cancers was classified as weak, based on 17 meta-analyses. The study also highlighted the effect of “moderate” alcohol intake; individuals who consumed 12.5 g to 50 g per day—the equivalent of 1 to 3.7 drinks per day—had a higher risk in comparison to non-drinkers, RR = 1.28 (1.10–1.49) [18].

Two prospective studies failed to prove any relation between alcohol consumption before the age of 23 years old and breast cancer risk [16,17]. However, one study focused on alcohol consumption during the interval between two important events of reproduction: onset of menstruation and first birth. The relationship between alcohol intake during this interval and the risk of breast cancer depends on the duration of the interval. Among women with a greater interval between menarche and first pregnancy (10 or more years), each 10 g/day increase of alcohol consumption increases the risk of breast cancer by 21%, regardless of alcohol intake after the first pregnancy. Among women with a shorter interval between menarche and first pregnancy, alcohol consumption does not increase the risk of breast cancer [15].

The limited data available demonstrate the link between alcohol consumption during adolescence and adulthood and the occurrence of BBD. Berkey et al., in a prospective cohort study, assessed drinking habits among females aged 16–23. In this analysis drinking 6 to 7 days per week was associated with a more than fivefold risk for BBD, whereas drinking 3 to 5 days per week was associated with a more than threefold risk in comparison to participants who consumed alcohol less than once a week or did not consume any at all [19]. Data from the study NHS II (Nurse’s Health Study II) show that the increasing levels of alcohol consumption before the first pregnancy, but not after it, proliferativelyincrease the risk for BBD [20].

The aforementioned conclusions, along with reports suggesting that adult alcohol intake does not elevate the BBD likelihood [21], indicate that early life alcohol intake has the greatest effect on these breast conditions. The effect of alcohol in adolescence and risk of BBD may be particularly strong for young females with a breast cancer family history or a maternal history of BBD [22].

Several mechanisms have been proposed for alcohol’s effects on the breast, but it is still unknown which are responsible for increased risk of BBD and breast cancer. Proposed mechanisms include an effect on circulating hormone levels, the production of carcinogens such as acetaldehyde, and oxidative stress [23].

#### 3.2.2. Physical Activity (PA)

Monninkof et al. found an inverse relationship between PA in adolescence and breast cancer in approximatelyhalf of the studies that assessed PA before the age of 20 [24]. Some studies have reported that recent PA has a stronger protective effect in comparison to PA of the distant past [25].

The positive impact of adolescent PA on breast malignancies prior or following menopause has been previously reported. PA, however, may need to be sustained until adulthood to retain its protective effect. In the Nurse’s Health Study II, for example, a reduced risk of premenopausal breast cancer was most apparent among women who engaged in high levels of activity during both youth (ages 12–22) and adulthood compared with women with low levels of activity during both age periods, as active women had a 30% reduction of breast cancer risk (RR = 0.70, 95% CL: 0.53–0.93) [26].

In a recent study by Boeke et al., PA between 14 and 22 years of age appeared as modestly protective for premenopausal breast cancer, but taking into consideration other parameters such as exercise in adulthood and BMI this correlation was reduced. Furthermore, the correlation appeared slightly stronger in younger premenopausal women with estrogen receptor-negative tumors, although the differences were not statistically significant [27].

There is little evidence suggesting that PA during childhood and adolescence has a significant impact on BBD development. Baer et al. detected a protective effect (RR: 65.95% CL: 50–84) in women who performed “strenuous” PA 4 to 6 months per year while they were in high school; however, this effect was not apparent in women who reported more frequent strenuous activity during this time period [28]. A recent analysis studied lifetime PA, including frequency, duration, and type, and reported an association between increased PA and reduced risk for P-BBD [29].

Several mechanisms may explain the protective effect of PA, and their function is clearer, especially during adolescence. Menarche often delays in adolescent athletes, and there are reduced menstrual cycles or anovulation, which changes sex hormone exposure. High levels of endogenous hormones such as estrogens, androgens, and prolactin increase breast cancer risk [30]. Physical activity can also reduce fat tissue and therefore change adipokine exposure [27]. Moreover, PA increases insulin sensitivity by reducing Insulin Growth Factor 1 (IGF-1) and inflammation, which may help to protect against breast cancer [30].

In a meta-analysis by Hidayat et al., which included eighty publications and was published in 2020, extensive PA, both at a younger age as well as later in life, is shown to be associated with a reduced incidence of breast cancer [31].

### 3.3. Anthropometric Factors

#### 3.3.1. Body Mass Index and Weight

According to Van den Brandt et al., there is a negative correlation between adult adiposity and premenopausal breast cancer and a positive correlation between adult adiposity and postmenopausal breast cancer [32]. In prospective studies, childhood and adolescent adiposity show a negative correlation with breast cancer during the postmenopausal years, even after controlling for adult attained weight or BMI [3]. Systematic reviews and meta-analyses show an inverse trend between late adolescent BMI and premenopausal breast cancer observed in Caucasian and African heritages, though evidence from Asia is more variable [33].

This inverse relation to premenopausal breast cancer, described above, is also observed for proliferative BBD. Berkey et al. found that higher BMI, as measured during adolescence, was associated with slightly decreased BBD risk. Girls with a BMI in the upper two quintiles of BMI had less than half the risk (OR: 46 95% CL: 26–81) compared with those with a BMI in the lower three quintiles [34]. This finding was consistent with results from the NHS II, supporting that body fat composition measured in children between 5 and 10 was inversely related to P-BBD risk. This protective effect was also apparent in later adolescence: a BMI> or equivalent to 25 at age 18 was associated with a 33% reduction in BBD risk [28].

Following the Nurses’ Health Study (NHS) and NHS II, where the rates of individual breast tissue types have been quantified, Oh et al. conducted a thorough study examining the relation of early-life and adult anthropometric measures with breast tissue composition on BBD among women with biopsy-confirmed BBD enrolled in the NHS and NHS II. Regarding the early life anthropometric measures, which is the main concern in this particular literature review, it appeared that the amount of body fat in children was associated with a lower proportion of fibrous stromal tissue. ΒΜΙ at age 18 was also significantly associated with adipose tissue and inversely associated with the proportion of epithelial and fibrous stromal tissue. Oh et al. [35] indicated that higher levels of adipose tissue in the body are related to lower levels of dense tissue (epithelial and fibrous stromal tissue) and higher levels of non-dense (adipose) tissue in the breast. It was also observed that an increase in the amount of adipose tissue in the breast may replace fibrous stromal tissue but not epithelial tissue [35].

#### 3.3.2. Growth Velocity and Height

Several studies suggest that a rapid height growth during puberty may constitute a factor for the development of cancer. When childhood growth is rapid, there is less time available for the repairment of DNA damage caused by exposures to carcinogenic factors [36].

A Danish study by Allgren et al., in which the annual height and weight of children was collected from school health records, reported that height growth from age 8 to 14 years was significantly associated with a high risk of developing breast cancer (RR: 1.17/5 cm increase), (CI 95%: 1.09–1.25), while growth during the peak year showed a marginally significant correlation (OR: 1.15/5 cm increase, CI: 0.97–1.36) [37]. In a British cohort study, it was found that rapid height growth from age 4 to 7 years, and from age 11 to 15 years, were associated with increased risk for breast cancer [38].

More recent evidence supports the link between height growth velocity and BBD. In the GUTS (Growing Up Today Study), Berkey et al. reported that a faster rate of growth was associated with a risk for BBD; girls with peak height velocity >8.9 cm/year were nearly twice as likely to develop BBD in comparison to girls whose peak height velocity was below or equivalent to 7.6/year [34].

Equally interesting, in 2020, the Sisters Study Cohort, a prospective cohort study of US women with a family history of breast cancer, examined all pubertal markers in the context of their family history. It a study by Goldberg et al., it was found that women who reached their adult height at 18 years of age or later had a 13% decreased risk of developing breast cancer compared to those reaching adult height at 15–17 years of age [39].

### 3.4. Age at Menarche

Earlier age at menarche is related to an increased risk of premenopausal and postmenopausal breast cancer. In a meta-analysis including more than 100 epidemiological studies, each one-year decrease in age at menarche increased the risk of breast cancer by 5% [40]. The underlying mechanisms are not well understood but may involve higher levels of estrogen both earlier [41] and later [42] in life in girls with earlier menarche.

In the Multiethnic Cohort Study, age at menarche was associated with positive estrogen receptor (ER+) and positive progesterone receptor (PR+) breast cancer, but not with ER−/PR− breast cancer [43]. In addition to any hormone-mediated effects, another potential mechanism by which early age at menarche could increase breast cancer risk is the lengthening of the interval between menarche and first birth [44].

Furthermore, in the Sister Study, the established risk factor of earlier age at menarche was confirmed and associated with an increased risk of breast cancer for <12 years compared to 12–13 years [39].

The relationship between age at menarche and risk of BBD is not yet well established [34]. In the GUTS cohort, Berkey et al. reported an absence of correlation between age of menarche and BBD risk [22,34]. Lack of correlation between the aforementioned variables was further confirmed by several other studies [45].

Tamimi et al. reported a differential impact of the pathologic subtype of BBD on the relationship between age at menarche and subsequent breast cancer risk among women in the NHS II [46]. A reduced likelihood of breast carcinoma (RR: 93.95% CI: 86–99) was reported in females with P-BBD without atypia. In contrast, females with non-proliferative BBD who were age 15 at menarche had a higher risk (RR: 1.16, 95% CI: 1.08–1.24) compared with women who were 11 at menarche [46].

Age at menarche is determined partially by hereditary factors, but body size, nutrition, and physical activity may also play a role [47]. Menarche tends to be earlier in girls with increased body fat and later in girls who exercise [48]. A childhood diet that is high in animal protein and low in vegetable protein may also be linked with earlier menarche [49].

### 3.5. Age at Thelarche

In the Sister Study cohort, Goldberg et al. thoroughly examined the age of menarche, as well as other pubertal factors in relation to breast cancer. In particular, the age of thelarche was shown to be biologically more relevant to breast cancer risk than the timing of menarche, as it represents the onset of the vulnerable period of rapid breast cell proliferation. Thelarche prior to 10 years of age was associated with a 23% greater risk of breast cancer compared with the mean age of thelarche at 12–13 years. A 3% decrease in breast cancer risk was associated with each 1-year delay in age at thelarche. The timing of thelarche was inversely associated with the risk of development both ER+ and ER− cancers [39].

The early onset of puberty appears to trigger the prolonged exposure of breast cells to a highly proliferative, undifferentiated state, making them more susceptible to carcinogenesis. Earlier puberty, caused by the re-activation of the hypothalamic-pituitary-gonadal axis, triggers the increase in endogenous hormones such as estrogen and insulin-like growth factor-1 (IGF-1), which regulate breast development [39].

### 3.6. Ionizing Radiation

Children who received therapy with chest radiation for other pediatric malignancies are known to be at increased risk of developing breast cancer later in life. Radiation exposure for girls during peak breast development, typically from 10 to 16 years of age, is associated with the highest risk. Approximately 40% of girls treated with radiation for Hodgkin lymphoma will develop breast cancer; it takes an average of 20 years for it to appear [50].

Breast cancer risk is greatest among women treated for Hodgkin’s lymphoma with high-dose mantle radiation, but it is also elevated among women who received moderate-dose chest radiation (e.g., mediastinal, lung) for other pediatric and young adult cancers, such as non-Hodgkin’s lymphoma, Wilms tumor, leukemia, bone cancer, neuroblastoma, and soft tissue sarcoma [51].

Inskip et al. reported that the risk of being diagnosed with breast cancer increased with chest radiation dose, reaching 10.8 (95% CI: 3.8–31) for 40 Gy compared to those who received no radiation [52]. In addition, a later study found that among women treated for childhood cancer with chest radiation therapy, those treated with whole-lung irradiation have a greater risk of breast cancer, demonstrating the importance of radiation volume [53].

Evidence has also suggested that the risk could be associated with radiation field volume, due to the increased risk (odds ratio 2.7 (95% CI, 1.1–6.9) of women treated with mantle field irradiation compared to women with similarly dosed mediastinal irradiation (omitting the axillary nodes) [54].

In a study by Schellong et al., up to July 2012, secondary breast cancer was diagnosed in 26 of 590 female patients with Hodgkin disease (HD). Their age at time of treatment for HD was 9.9 to 16.2 years. Radiation to the supradiaphragmatic fields was between 20 and Gy. The cumulative incidence for secondary breast disease 30 years after treatment for HD was 19% (95% CI, 12% to 29%). Women aged 25 to 45 reported a frequency of breast cancer 24 times as high as in the corresponding normal population [55].

Given that radiation therapy is a confirmed risk factor for a second breast cancer, Charlotte Demoor-Goldschmidt et al. characterized for the first time the histological subtypes and the hormonal receptor status of radiation therapy-induced SBC among survivors of a childhood or young adult cancer. In particular, a multicenter retrospective study of 121 women was conducted, with the mean age of the first cancer diagnosis at 15 years and at initial SBC diagnosis at 38 years of age. The main finding of the study associated radiation doses greater than 20 Gy to the mediastinum with triple negative phenotype breast cancers, which in turn were related to an aggressive histoprognostic status [56].

In one of the largest studies of treatment-related breast cancer following childhood cancer, Veiga LH et al. examined breast cancer risk according to radiation dose to the breast and ovaries. Veiga et al. reported the proportional relationship between the overall risk of developing breast cancer and the radiation dose to breast cancer location and the inverse relationship between the overall risk of breast cancer and the radiation dose to ovaries. Specifically, the odds ratio (OR) in a radiation dose of 10 Gy was 3.9, and it was significantly elevated even for an irradiation dose lower than 5 Gy [57].

In the same study, high doses to the ovaries (>=15 Gy) were related to lower odds of breast cancer associated with the radiation to the breast, while women receiving less than 1 Gy to the ovaries faced higher breast cancer risk. For both ER+ and ER– invasive breast cancers, the protective role of ovarian radiation was reported [57].

Several other studies, including the Late Effects Study Group (LESG) cohort study, considered survivors of Hodgkin lymphoma, who were diagnosed and were treated during childhood, as a high-risk group for developing subsequent malignant neoplasms. On this basis, Holmqvist et al. pointed out the elevated risk of breast cancer in female survivors, and specifically it was associated with a 25.8-fold increased risk when compared to the general population. According to the study, the patients who face the highest risk of developing breast cancer by the age 50 years are those who were diagnosed with HL between ages 10 and 16 years, those treated with chest radiotherapy, and those who had received no or a very low dose of alkylating agents. Consistent with this finding is the recommended early screening at the age of 25 years or after 8 years from the time of HL diagnosis that would benefit this group of survivors after HL [58].

### 3.7. Anthracyclines

In 2016, Henderson et al. attempted to identify the risk factors for breast cancer among childhood cancer survivors who did not receive chest radiotherapy. Henderson et al. conducted the Childhood Cancer Survivor Study, in which 3768 females without a history of chest radio therapy as part of their cancer treatment plan during childhood participated. The development of subsequent breast cancer, on the ground of nonirradiated field, was associated with exposure to anthracyclines or alkylators. Henderson et al. reported a 2.5- to 6-fold increased risk of breast cancer in women received anthracycline chemotherapies. The findings of the study confirmed the potential of anthracycline agents to trigger malignant transformations in mammalian cell systems as well as the ability of alkylators to disrupt cancer growth or cause DNA damage [59].

Following the previous study, in 2019Ehrhardt et al. tried to clarify the relation between anthracycline-associated risk for the development of subsequent breast cancer and TP-53 mutation-related gene-environment interactions. In the context of this cohort study, in which the participants were 1467 childhood cancer survivors who were treated at St Jude Children’s Research Hospital, a greater than 13-fold risk for breast cancer in women who received 250 mg/m^2^ or more of anthracycline was observed. Furthermore, breast cancer predisposition gene mutations were identified, and it was suggested that the anthracycline treatment-related risk was independent of autosomal dominantly inherited cancer predisposition mutations, especially for TP53 [60].

### 3.8. Socioeconomic Status

In 2017, Hiatt et al. described the impact of socioeconomic position (SEP) on pubertal onset. In more detail, puberty onset was assessed based on the standard methods of Tanner staging and defined by observation and palpation of breast budding for stages B2 or higher as well as by observation of stages PH2 and higher.

The main finding of this prospective cohort study was that girls in the lowest SEP index quantile tend to develop pubertal signs of breast budding 7 months earlier than girls in the highest SEP quantile. Given that earlier onset of female reproductive maturity is associated with increased breast cancer rates in adulthood, SEP could be indirectly related to breast pathologies. SEP, therefore, may influence breast and generally pubertal development onset [61].

### 3.9. Smoking and Drug Abuse

With regard to BBD, there is limited data available. Cui et al. reported no increased risk of postmenopausal, benign proliferative epithelial disorder (BPED) among women who started smoking during adolescence [62]. Regarding breast cancer risk, a review by Okasha et al. published in 2003 concluded that the data supporting an association between smoking at a young age and breast cancer risk is inconsistent, and the same applied for passive smoking in early life, highlighting the need for further studies to evaluate this relationship [63]. However, a more recent study showed the opposite; a study by Jones et al. concluded that smoking was associated with an increased risk of breast cancer, particularly in women who began smoking during adolescence as well as in women with a family history of the disease [64]. Interestingly enough, a study by Liu et al. demonstrated that exposure to heavy cigarette smoking during the prenatal period could be associated with an increased risk of BBD in adulthood [65].

### 3.10. Other Factors

Limited data have suggested an association between drug and antibiotic abuse during late adolescence and breast cancer. A study by Dahlman et al., including 3,838,248 women aged 15–75 years in Sweden, concluded that women with drug use disorders constitute a risk group for incident, fatal, and metastasized breast cancer [66].Furthermore, a study by Velicer et al., including patients who were at least 19 years old, showed that cumulative days of antibiotic use, especially as a treatment for respiratory tract infections, constitutes a risk factor for the development of breast cancer [67]. However, both studies could not determine if the medication use was causally related to breast cancer. This has also been confirmed by a recently published meta-analysis by Simin et al., highlighting the need for further studies to examine this relationship [68].Evidence has also highlighted the anticancer properties of Vitamin D in relation to breast cancer [69]. Researchers hypothesize that this could explain the seasonality of the diagnosis, with the highest diagnosis rates during spring and autumn, as solar ultraviolet radiation B through the production of vitamin D lowers the risk for the disease in summer and higher concentrations of melatonin reduce the risk during winter [70]. Interestingly, serum 25(OH)D concentrations have also been associated with the risk for breast cancer; the risk increased rapidly as serum concentrations decreased below 12 ng/mL [69]. Lastly, Vitamin D from sunlight exposure has been associated with a lower risk of fatal breast cancer [71].

## 4. Discussion

In this review, we examined the impact of various factors, such as diet and exercise, obesity, and age at menarche on the development of breast disorders. Taking into consideration that their impact begins from early childhood and adolescence, their identification will lead to the understanding of the pathogenesis of breast disease later in adulthood (Table 1) (Figure 1). Childhood and adolescence are very sensitive time periods for the development of breast pathology because breast tissue undergoes rapid proliferation during the time period between menarche and first full-term pregnancy. Overall, we draw the conclusion that a greater meat consumption during adolescence may be associated with an increased risk for breast cancer. Similarly, adolescents who consumed higher levels of fiber had a lower risk for proliferative benign breast diseases. Alcohol consumption also seems to be associated with a higher risk for breast cancer. The same correlation was found for benign breast disease. Physical activity during adolescence was related to a reduced risk for breast cancer (BC). Interestingly, some studies proposed that as the body mass index increases, there is a reduction in the premenopausal BC. The same inverse relationship is observed for P-BBD. On the other hand, growth velocity during childhood and adolescence was associated with statistically significant risk of BC and BBD. Women exposed to chest radiation due to various malignancies as well as cancer survivors who received anthracyclines in childhood and adolescence are at increased risk for BC. Vitamin D seems to have a protective effect against the development of breast cancer. Lastly, SEP is mentioned as a risk factor for the early onset of puberty. To our knowledge, this is one of the very few papers that critically review the factors that act during childhood and adolescence and could potentially be associated with the development of breast disease. The limitations of this review include the great heterogeneity of the studies, as the age of the sample as well as the number of participants differed greatly, as well as the lack of evidence to support certain associations as presented above. Furthermore, the small number of cases and the fact that data extracted by questionnaires combined with the retrospective collection of lifestyle risk information based on participant’s recall of adolescent risk factors are often confounded by ascertainment and recall bias. Lastly the lack of complete follow up could have resulted in either overestimation or underestimation of the risk, and also missing data such as exposure in passive smoking, diet, undetermined indications of use of antibiotics, etc., would be another potential weakness of our study. Another limitation concerns the fact that assay methods and cut offs such as anthropometric measurements and hormones varied between laboratories, registries, and across populations and studies, as well as different detection methods of breast cancer such as mammography and biopsy. In addition, a language criterion was applied, resulting in the exclusion of potentially relevant papers.

Our review identified gaps in knowledge that need to be covered in order to organize effective prevention strategies. Because there is growing evidence supporting the idea that parameters such as childhood diet, growth in height, and adolescent alcohol intake are related to breast cancer risk and risk of premalignant proliferative lesions, breast cancer prevention efforts should be initiated at an early age [44,72]. It has been suggested that lifestyle modifications among high-risk women, those for example with a positive family history and those who are carriers of BRCA mutations, could prevent 25% to 30% of cases of breast cancer [73]. This can be achieved if targeted prevention and screening programs are implemented during childhood and adolescence, periods during which the developing breast is particularly susceptible to carcinogenesis.

## 5. Conclusions

The available literature provides clear evidence that several nutritional, lifestyle, iatrogenic, and socioeconomic risk factors, already presenting during childhood and adolescence, may significantly influence the development of various benign and malignant pathologies of the breast. However, the need for more robust evidence from prospective and retrospective studies investigating their long-term impact is evident. It is also high time to sensibilize gynecologists and adolescent care specialists and raise awareness regarding the prevention of breast disease, and therefore appropriate guidance and interventions are necessary from early life and should be part of standard care.

## Figures and Tables

**Figure 1 children-10-00601-f001:**
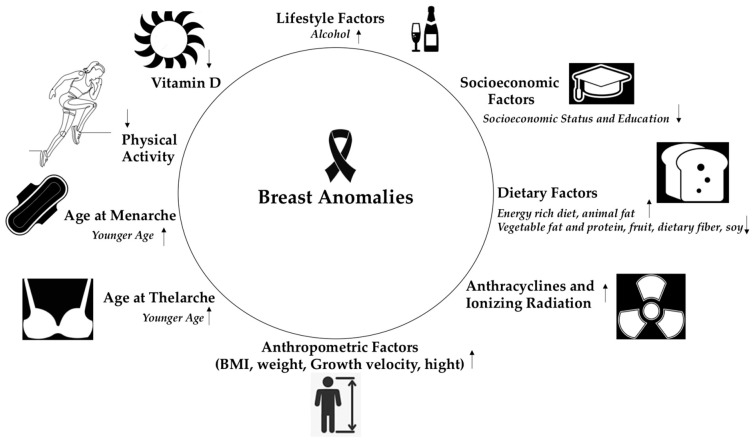
Factors during childhood and adolescence that interfere with the development of proliferative benign breast diseases and breast cancer in adulthood; 
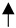
: the risk increases, 
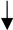
: the risk decreases.

**Table 1 children-10-00601-t001:** Factors during childhood and adolescence that interfere with the development of proliferative benign breast diseases and breast cancer in adulthood.

Factors	Effect	Reference
**Dietary factors**	An energy-rich diet is associated with an increased occurrence of BBD	Trichopoulos [4]
	Animal fat consumption increases the risk of P-BBD High vegetable fat consumption reduces the risk of P-BBD	Berkeley et al. [5], Baer et al. [6], Linos et al. [7]
	Fruit consumption is associated with a lower risk of breast cancer	Farvid et al. [8]
	Dietary fiber and vegetable protein consumption reduces the risk for breast cancer	Liu et al. [10]
	Soy intake reduces the risk for breast cancer	Lee et al. [13]
Alcohol	Few studies available, limited data demonstrate the link between alcohol consumption and the occurrence of BBD and breast cancer	Berkey et al. [18]
Physical activity	Limited data regarding BBD Has a protective effect against breast cancer	Maruti et al. [26]
BMI and weight	A BMI> or equivalent to 25 at age 18 was associated with a reduction in BBD risk	Baer et al. [28]
Growth velocity and height	A faster height velocity was associated with a higher risk for BBD and breast cancer	Berkey et al. [34], Ahlgren et al. [37], De Stavola et al. [38]
Age at Menarche	Each one-year decrease in age at menarche increased the risk of breast cancer by 5%	Collaborative Group, The lancet oncology [40]
	The relationship between age at menarche and risk of BBD is not yet well established	
Age at Thelarche	Each one-year delay in age at thelarche was associated with a 3% decrease of breast cancer risk	Goldberg et al. [39]
	The relationship between age at thelarche and risk of BBD is not yet well established	
Ionizing Radiation	Chest radiation for pediatric malignancies increases the risk for breast cancer	Henderson et al. [51]
Anthracyclines	Exposure to anthracyclines or alkylators increases the risk for breast cancer	Henderson et al. [59]
Socioeconomic Status	SEP could be indirectly related to breast pathologies	Hiatt et al. [61]
Smoking	Few studies available, conflicting data	Cui et al. [62]
Vitamin D	Anticancer properties	Muñoz et al. [69]

## Data Availability

Not applicable.

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
