# Peer review of "Investigating the Link between Early Life and Breast Anomalies"

_children, 2023, doi:10.3390/children10030601_

Round 1

Reviewer 1 Report

Dear authors,

Thank you for submitting your manuscript to this journal. The topic is really very exciting and should get more attention in the future.

You have listed some points that may be responsible for the development of breast carcinoma and have cited many manuscripts on this.
I personally miss a graphical or tabular summary especially for the given topics. This would improve the overview and makes it easier to see your research once again.

Furthermore, a subdivision into methods and results is missing, perhaps this is not required by the editor, this should be evaluated again.
In the abstract you describe your literature research, but in the further review these are no longer mentioned. I would supplement this, also in which period you have researched.

Furthermore, the results could still be discussed. Were there limitations in some studies?
Can the mentioned studies be compared well at all (number of patients, age...)?

I hope you can improve your review with these suggestions and wish you continued success.

Reviewer 2 Report

In this manuscript, Christopoulos et al. reviewed the current evidence regarding the link between early life and breast anomalies, and discuss the challenges and difficulties in the management of these patients. The authors considered that  diet, exercise, age of menarche, body mass index, ionizing radiation exposure during childhood and adolescence as the risk factors of proliferative breast diseases and breast cancer in adulthood. They concluded that numerous studies concede that the development of breast disease in adulthood is influenced by various risk factors, whose influence begins during early childhood and adolescence, but it remains many problems which still need to be challenging. This review is important for breast disease prevention. There are several concerns with the study:

1. The authors should add a Figure or a Table to summarize the evidence from the published literature.

2. Are there other lifestyle factors during childhood and adolescence that contribute to breast disease? Such as smoking, school bullying, stay up late, and so on.

3. Also, the authors should discuss whether drug abuse (e.g. antibiotic) during childhood and adolescence could contribute to breast disease.

4. Are there ways to avoid these risk factors during adolescence to prevent breast cancer? What are the difficulties and challenges? Please list in the appropriate section.

5. There are still some grammatical and spelling errors throughout.

Round 2

Reviewer 2 Report

The revised manuscript has made a great improvement. I have no more comments and recommends.

Author Response

Thank you for your kindness to add your positive comments.